# NaviTerm Scientific Article Search : Un outil de recommandation séquentielle d'article scientifique basé sur l'historique de lecture

Léane Jourdan[1]    Julien Aubert-Béduchaud[1]    Florian Boudin[2]    Beatrice Daille[1]
Richard Dufour[1]

(1) Nantes Université, École Centrale Nantes, CNRS, LS2N, UMR 6004, F-44000 Nantes, France
(2) Inria, LS2N, Nantes Université, France
`prenom.nom@univ-nantes.fr`

RÉSUMÉ

La recherche d'articles scientifiques repose majoritairement sur des requêtes explicites et des recommandations peu sensibles au contexte de lecture. Nous présentons *Naviterm Scientific Article Search*, un outil de démonstration dédié à la recommandation séquentielle d'articles scientifiques à partir de l'historique de lecture des utilisateurs. Le système modélise un profil utilisateur dynamique en combinant la requête initiale et les articles consultés, représentés sous forme de plongements obtenus avec SPECTER2. Les recommandations sont générées par similarité vectorielle au sein de la collection ACL Anthology. L'interface interactive permet d'explorer les résultats, de filtrer les articles et de simuler des parcours de lecture. *Naviterm* met ainsi en évidence l'intérêt des approches séquentielles pour la découverte de littérature scientifique dans des scénarios d'exploration progressive.

ABSTRACT

**NaviTerm Scientific Article Search : A tool for sequentially recommending scientific articles based on reading history**

Scientific article search systems mainly rely on explicit queries and often provide recommendations that are weakly adapted to the user's reading context. We present *Naviterm Scientific Article Search*, a demonstration tool for sequential recommendation of scientific articles based on user reading history. The system models a dynamic user profile by combining the initial query and previously read articles, represented using SPECTER2 embeddings. Recommendations are generated through embedding similarity over the ACL Anthology collection. The interactive interface enables users to explore results, apply filters, and simulate reading trajectories. *Naviterm* highlights the benefits of sequential recommendation approaches for scientific literature exploration in progressive discovery scenarios.

MOTS-CLÉS : démo, articles scientifiques, moteur de recherche, recommandation séquentielle.

KEYWORDS: demo, scientific articles, search engine, sequential recommendation.

## 1    Introduction

Les moteurs de recherche académiques tels que *Google Scholar*[1] ou *Semantic Scholar*[2] sont devenus des outils incontournables pour l'accès à la littérature scientifique. Ils permettent une recherche

---

1. `https://scholar.google.com/`
2. `https://www.semanticscholar.org/`

efficace à partir de mots-clés et offrent des fonctionnalités de tri, de filtrage ou de recommandation. Toutefois, ces systèmes présentent certaines limites : ils tendent à décontextualiser les travaux par rapport à la littérature existante dans le domaine, en privilégiant la pertinence des résultats vis-à-vis des termes de la requête formulée par l'utilisateur.

Pour pallier cette limitation, la génération de listes de lecture scientifique (Ekstrand *et al.*, 2010; Aubert-Béduchaud *et al.*, 2025) vise à recommander des articles afin de guider les chercheurs, en particulier les novices, dans un domaine donné. La majorité des approches existantes se concentre sur la sélection des contenus (*quoi lire*), produisant des listes d'articles ciblées sur un domaine précis (Ekstrand *et al.*, 2010; Sesagiri Raamkumar *et al.*, 2017). D'autres travaux récents se sont intéressés à l'organisation des contenus (*comment lire*), en structurant les articles selon les dépendances de connaissances ou l'évolution du domaine (Gordon *et al.*, 2017; Ding *et al.*, 2022). Néanmoins, ces méthodes génèrent généralement des listes statiques, sans tenir compte des connaissances déjà acquises par l'utilisateur.

D'autres travaux dans le domaine de la recommandation se sont principalement intéressés à la suggestion de produits à partir de l'historique d'interactions d'un utilisateur (Kang & McAuley, 2018; Sun *et al.*, 2019; Boka *et al.*, 2024). À notre connaissance, ces méthodes n'ont pas encore été adaptées à la recommandation d'articles scientifiques.

Dans ce contexte, nous présentons *NaviTerm Scientific Article Search* (*NaviTerm*), un outil qui vise à faciliter l'exploration de collections d'articles scientifiques en proposant un cadre de recommandation séquentielle à partir de l'historique de lecture des utilisateurs.

Plus précisément, *NaviTerm* propose une interface interactive de navigation au sein d'une collection d'articles scientifiques, en l'occurrence l'ACL Anthology, permettant de simuler des parcours de lecture et d'explorer des recommandations personnalisées au fil de l'interaction. À chaque étape, le système met à jour l'historique de lecture en fonction des articles consultés et génère de nouvelles suggestions adaptées au contexte courant.

La méthode de recommandation repose sur le calcul de similarité entre agrégations de plongements. Le profil utilisateur est modélisé par une agrégation des plongements associés à la requête initiale et aux articles précédemment consultés. Ce profil est ensuite comparé aux plongements des articles candidats afin de produire un classement pertinent. Les représentations des articles sont construites à partir de leur titre et de leur résumé, capturant ainsi des informations sémantiques essentielles.

D'un point de vue technique, l'architecture de l'outil repose sur un frontend développé avec Vue.js/Vite et un backend implémenté en FastAPI. L'outil est accessible en ligne à l'adresse `naviterm.ls2n.fr`, et son code source est disponible publiquement.[3]

## 2 Recommandation séquentielle d'articles scientifiques

**Définition de la tâche**  La recommandation d'articles scientifiques dans les systèmes traditionnels repose sur la modélisation d'un besoin d'information exprimé par l'utilisateur (à travers des mots-clés, des requêtes ou des questions), afin d'identifier et de proposer des articles pertinents au sein d'une collection, en fonction de leur similarité avec ce besoin.

Afin de personnaliser l'expérience utilisateur, nous proposons une nouvelle tâche de recommandation

---

3. https://github.com/JourdanL/demo_naviterm

séquentielle adaptée à la recommandation d'articles scientifiques (voir Figure 1).

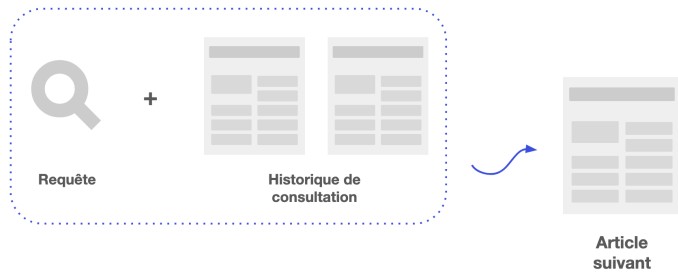

FIGURE 1 – Tâche de recommandation séquentielle d'articles scientifiques

Soit $D$ un domaine de recherche, $Q_d$ une requête initiale liée à ce domaine formulée par un utilisateur, et $H_d = (a_1, \ldots, a_n)$ avec $a_i \in D$ son historique de lecture. La recommandation séquentielle vise à prédire le prochain article pertinent en combinant ces deux signaux :

$$a_{n+1}^* = \arg\max_{a \in D} \ P(a \mid Q_d, H_d).$$

**Collection d'articles scientifiques**  Nous utilisons la collection d'articles scientifiques mise à disposition par l'ACL Anthology [4]. La version des données utilisée compte les méta-données de 118 325 articles provenant de diverses conférences du domaine du TAL (titre, résumé, année, auteurs, conférence, etc.).

Afin de permettre de faciliter la recherche de l'utilisateur et permettre de filtrer les résultats, nous assignons aux articles un ou plusieurs types de contribution réalisés. Nous étiquetons les types de contributions présentés dans chaque article à l'aide du modèle ContriBERT (Aubert-Béduchaud *et al.*, 2026a)[5], entraîné sur le jeu de données ARRContributions[6], qui propose des annotations auteurs des types de contributions d'articles soumis lors du processus ACL Rolling Review. Les types de contributions possibles sont au nombre de 11 *(Approaches to low-resource settings, Approaches low compute settings-efficiency*, *Data resources*, *Data analysis*, *Model analysis & interpretability*, *NLP engineering experiment*, *Publicly available software and/or pre-trained models*, *Position papers*, *Reproduction study*, *Surveys* et *Theory*).

**Modèle de recommandation**  Nous utilisons le modèle SPECTER2 (Singh *et al.*, 2022)[7] pour réaliser les plongements lexicaux des articles et des requêtes utilisateur. La prédiction de l'article suivant consiste en une interpolation du besoin d'information et de la moyenne du contenu de l'historique de lecture.

Soit le vecteur $\hat{q}$ permettant de requêter la collection, obtenue par interpolation entre le plongement du besoin d'information $\hat{Q}_d$ et la moyenne des plongements de l'historique $\hat{H}_d = \frac{1}{n}\sum_{i=1}^{n}\hat{a}_i$, contrôlée par un paramètre $\alpha \in [0,1]$ (par défaut, $\alpha = 0.5$) :

---

4. https://pypi.org/project/acl-anthology/
5. https://huggingface.co/taln-ls2n/ContriBERT-ACL
6. https://huggingface.co/datasets/taln-ls2n/ARRContributions
7. https://github.com/allenai/SPECTER2

$$\hat{q} = \alpha \, \hat{H}_d + (1 - \alpha) \, \hat{Q}_d, \qquad a_{n+1}^* = \arg\max_{a \in D} \, \hat{q}^\top a$$

Le protocole d'évaluation détaillé du modèle ainsi que les différentes mesures d'évaluation utilisées sont présentés dans Aubert-Béduchaud *et al.* (2026b).

# 3 Interface

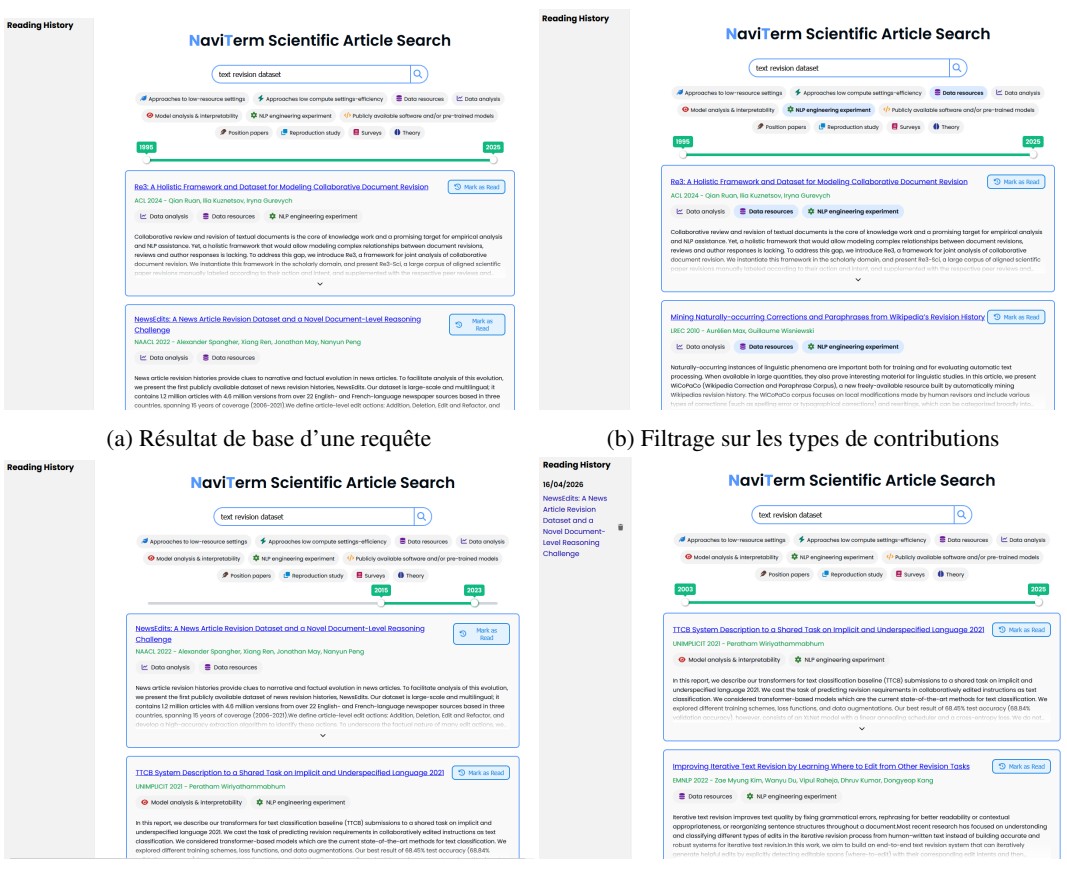

(a) Résultat de base d'une requête    (b) Filtrage sur les types de contributions

(c) Filtrage sur la date de publication    (d) Ajout d'un article à l'historique de lecture

FIGURE 2 – Captures d'écran de l'interface de Naviterm

L'interface de *NaviTerm* s'inspire des moteurs de recherche académiques classiques tels que *Google Scholar*, tout en intégrant des fonctionnalités dédiées à la navigation séquentielle. L'outil propose une barre de recherche permettant à l'utilisateur de saisir des mots-clés ou une requête en langage naturel. Sous cette barre, deux mécanismes de filtrage, par type de contribution et par date, sont disponibles afin de raffiner les résultats. Par ailleurs, un panneau latéral situé à gauche de l'écran affiche l'historique de lecture.

À l'issue d'une recherche, les articles sont présentés sous forme de liste, voir un exemple en Figure 2a. Pour chaque entrée, les informations principales sont affichées : titre, auteurs et conférence de publication. Le résumé (*abstract*) est présenté de manière tronquée afin de préserver la lisibilité, avec la possibilité de le déployer à la demande. Le titre de chaque article constitue un lien cliquable redirigeant vers sa page correspondante dans l'ACL Anthology.

**Filtrage par type de contribution**     Chaque article est associé à un ou plusieurs types de contribution (entre un et trois). Ces catégories sont accessibles sous forme de filtres situés sous la barre de recherche. L'activation d'un filtre permet de restreindre l'affichage aux articles correspondant aux catégories sélectionnées (exemple en Figure 2b). Il est possible de combiner plusieurs filtres afin d'affiner davantage les résultats. Ce mécanisme de filtrage s'applique uniquement aux résultats déjà retournés, sans relancer le processus de recherche.

**Filtrage par date**     Un second mécanisme de filtrage repose sur une sélection temporelle. Un curseur à double extrémité permet de définir un intervalle de dates dans lequel les articles doivent être compris (exemple en Figure 2c). Comme pour le filtrage par contribution, cette opération agit uniquement sur l'ensemble de résultats courant, sans déclencher une nouvelle requête.

**Gestion de l'historique de lecture**     La fonctionnalité centrale de *NaviTerm* réside dans la gestion de l'historique de lecture. Chaque article est associé à un bouton *Mark as read*, permettant de l'ajouter à cet historique. Cette action déclenche une mise à jour de l'historique et relance le processus de recommandation, produisant ainsi une nouvelle liste d'articles adaptée au contexte enrichi. Les articles sélectionnés apparaissent alors dans le panneau latéral dédié, un exemple est disponible en Figure 2d.

Il est également possible de retirer un article de l'historique via une icône de suppression associée. Cette opération entraîne une nouvelle mise à jour de l'historique et un recalcul des recommandations, garantissant la cohérence des résultats affichés avec l'état courant de l'historique.

# 4   Conception

L'outil repose sur une architecture client–serveur dans laquelle le frontend gère l'interaction utilisateur tandis que le backend prend en charge la logique de recommandation et l'accès aux données.

**Frontend**     Le frontend est développé avec Vue.js et Vite [8], offrant une interface réactive et fluide adaptée à une utilisation interactive en démonstration. Il prend en charge la saisie des requêtes, l'affichage des résultats, ainsi que la gestion de l'historique de lecture (en local storage), l'application des filtres. Cette architecture permet une mise à jour dynamique des recommandations en fonction des interactions.

**Backend**     Le backend est implémenté avec FastAPI [9], un framework Python dédié au développement d'API performantes et légères.

---

8. https://vuejs.org/
9. https://fastapi.tiangolo.com/

Le système reçoit en entrée une requête utilisateur ainsi que l'historique de lecture, représenté sous la forme d'une liste d'identifiants d'articles. La requête est ensuite transformée en plongement à l'aide de la bibliothèque Hugging Face, en s'appuyant sur le modèle SPECTER2, spécifiquement conçu pour la représentation de documents scientifiques. Les plongements des articles, construits à partir de leurs titres et résumés, sont pré-calculés et stockés en amont. Les identifiants des articles lus permettent de récupérer ces plongements. Ce choix permet de réduire significativement les temps de calcul lors des requêtes.

La liste des articles triés par pertinence de lecture est obtenue en calculant le score de chaque article non lu selon la méthode décrite en Section 2, afin de produire un classement des documents les plus pertinents. Les 100 meilleurs candidats sont conservés et retournés au frontend, permettant ainsi de limiter la charge computationnelle tout en garantissant une expérience utilisateur fluide.

# 5    Conclusion

Nous avons présenté *NaviTerm*, un outil de démonstration dédié à la recommandation séquentielle d'articles scientifiques, mettant à profit l'historique de lecture pour adapter dynamiquement les résultats pertinents. Des perspectives incluent l'intégration de fonctionnalité supplémentaires (multiples historique par thématique, comptes utilisateurs) ainsi que l'ouverture à une plus large collection de documents tel qu'ArXiv.

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
