# OpenReview forum: "NaviTerm Scientific Article Search: A tool for sequentially recommending scientific articles based on reading history"
_ls2n.fr/CORIA-TALN/2026/Workshop/ARTS — ls2n CORIATALN 2026 Workshop ARTS Submission_

### Official Review · Reviewer_vuGx · 2026-05-05

**Mode De Presentation:** Poster

**Confience:**

Oui

**Decision:**

Accepté

**Relecture:**

L'article est clair et facile à lire. Il présente bien le système
NaviTerm, notamment en terme de fonctionnalités.

Certains points pourraient être améliorés :

- Il n'est pas évident de savoir quel modèle est utilisé pour réaliser
  les plongements des articles de la collection. Est-ce qu'il s'agit
  de SPECTER2 ou ContriBERT ?

  Dans quelle mesure le système est dépendant de ces modèles ?

- Il manque un état de l'art même court. En quoi le système
  diffère-t-il des affectations d'articles dans OpenReview par exemple
  (comme "Open Recommendations") ?

- Comment est défini le paramètre $\alpha$ dans le vecteur $\circ{q}$ ?

- Il serait intéressant de présenter les performances du
  système sur la collection ACL Anthology, mais aussi les efforts
  nécessaires pour passer sur une autre collection, ou pour changer de modèle.

  De même, le coût, en terme de calcul ou de mémoire en fonction de la
  taille de la collection et de l'historique serait intéressant à présenter.

Typo :

page 3: "annotations auteurs" -> "annotations par les auteurs" (?)
        "obtenue" -> "obtenu"

**Resume:**

L'article présente un système de recommandation d'articles
scientifiques en tenant compte de l'historique de lecture. Le système
dispose d'une interface d'exploration des résultats et de possibilité
de filtrage.

Les articles précédemment consultés et les requêtes de l'utilisateur
sont représentés sous forme de plongements lexicaux proposés par
SPECTER2. Ces plongements sont ensuite agrégés pour modéliser le
profil utilisateur. Ce profil est ensuite comparé aux plongements des
articles présents dans une collection de documents (ici ACL Anthology).

---

### Official Review · Reviewer_SeD8 · 2026-05-05

**Mode De Presentation:** Poster

**Confience:**

Oui

**Decision:**

Accepté

**Relecture:**

L'article a tout à fait sa place dans l'atelier de par sa thématique (recommandation d'articles scientifiques). La prise en compte de l'historique de lecture dans la recommandation d'articles scientifiques est une proposition tout à fait intéressante. Par ailleurs, l'article présente un outil de démonstration opérationnel. Il y a néanmoins plusieurs axes d'amélioration: 1) l'article ne contient aucune évaluation de l'outil, 2) la bibliographie est très limitée avec uniquement deux références dont une référence anonyme (c'est assez surprenant car la littérature concernant la recherche d'articles scientifiques est volumineuse)

**Resume:**

L'article présente un nouvel outil de démonstration (client-serveur avec interface) qui permet de recommander des articles scientifiques à partir d'une requête en langue naturelle et d'un historique de lecture des articles. Pour le calcul des articles pertinents par rapport à la requête, les auteurs s'appuient sur le calcul de similarité entre les représentations vectorielles des articles et celle de la recherche (requête + historique de lecture) construits à partir de plongements lexicaux.

---

### Decision · Program_Chairs · 2026-05-07

Accept (Poster)